# TOWARDS MULTIMODAL UNDERSTANDING OF MUSIC SCORES AND PERFORMANCE AUDIO

## ABSTRACT

Music theory, scores, and performance audio are central modalities in music research, carrying rich information about melody, harmony, rhythm, and expressive interpretation. Yet, current multimodal large language models (MLLMs) struggle to reason jointly over symbolic and acoustic inputs, particularly when dealing with high-resolution scores and fine-grained performance signals. We introduce MuseBench, the first benchmark designed to evaluate MLLMs across three key dimensions of music understanding: (1) fundamental theory knowledge, (2) score-based reasoning, and (3) performance-level interpretation. To address these challenges, we further present MuseAgent, a multimodal retrieval-augmented large language model framework. MuseAgent employs two specialized perceptual modules: measure-wise optical music recognition (M-OMR) for sheet images and automatic music transcription (AMT) for performance audio. These modules unify heterogeneous modalities into structured textual representations (e.g., ABC notation, MusicXML, JSON), which can then be directly consumed by an LLM. A database retrieval module enables both explicit retrieval (user-driven) and implicit retrieval (agent-triggered) from symbolic and audio libraries, while also serving as a storage layer for structured music. Combined with a lightweight memory bank, MuseAgent supports multi-turn, interactive orchestration of modules according to user intent. Extensive evaluations on MuseBench show that MuseAgent substantially outperforms general-purpose MLLMs in symbolic and performance-level reasoning, demonstrating the effectiveness of combining structured multimodal representations, retrieval/storage, and agent-based orchestration.

## 1 INTRODUCTION

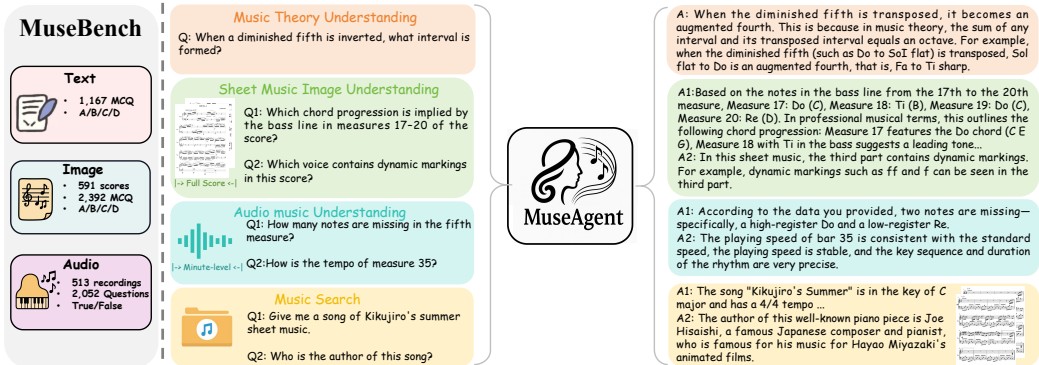

Figure 1: Overview of **MuseBench** and **MuseAgent**. MuseBench consists of multimodal music understanding tasks across text, image, and audio modalities, covering music theory, sheet score analysis, performance interpretation. MuseAgent integrates these modalities via sheet symbolic recognition, audio alignment, and retrieval modules, enabling large language models to answer complex music questions.

*"O Muses, O high genius, now help me!"*
— Dante, *Inferno*, Canto II, line 7

Music is a structured yet expressive domain, making it a compelling testbed for artificial intelligence. It spans symbolic representations such as notated scores and expressive acoustic outputs such as performances—each requiring distinct perceptual and reasoning abilities. As both a formal system and an emotional medium, music challenges AI models to reason across modalities with precision and nuance Essid & Richard (2012); Corrêard et al. (2021).

Among musical forms, piano music holds a uniquely central role in Western repertoire, with centuries of notation tradition and rich performance archives. The piano's wide pitch range, polyphonic texture, and use of both hands make it an ideal candidate for studying the intersection of symbolic and auditory modalities Hawthorne et al. (2019a). Understanding piano music requires bridging complex notational structures—pitch, rhythm, dynamics—with expressive features such as timing, articulation, and rubato. Such integration is critical for applications including transcription, accompaniment, digital archiving, and music education Benetos et al. (2019).

While prior work has advanced single-modality tasks—such as Optical Music Recognition (OMR) for scores and Automatic Music Transcription (AMT) for audio—integrated reasoning across modalities remains underexplored. Datasets like MAESTRO Hawthorne et al. (2019a) and generative models such as MusicLM Agostinelli et al. (2023) demonstrate the potential of symbolic and acoustic systems, but fall short of comprehensive multimodal understanding. Recent Multimodal Large Language Models (MLLMs), including GPT-4o OpenAI (2023) and Gemini Google Gemini Team (2023), promise cross-modal capabilities, yet they perform poorly on fine-grained music tasks due to limited domain-specific grounding and incomplete modality coverage. Systems such as AudioGPT Huang et al. (2024) and MusicAgent Yu et al. (2023) have begun coupling LLMs with domain-specific tools, but lack systematic benchmarks and often struggle with complex notation and long-form performance recordings. Recent studies on Retrieval-Augmented Generation (RAG) Lewis et al. (2020); Guu et al. (2020); Borgeaud et al. (2022) show that grounding large language models with external structured knowledge mitigates hallucination and enhances domain-specific reasoning. However, existing multimodal RAG approaches Shuster et al. (2022); Luo et al. (2023); Liu et al. (2023b) focus on text–vision–speech domains and rarely address music, where symbolic and acoustic modalities must be precisely aligned at high temporal resolution.

To address this gap, we introduce MuseBench, the first benchmark to evaluate MLLMs on joint understanding of music scores and performance audio. Centered on piano repertoire, MuseBench includes tasks such as score–audio alignment, performance error detection, and expressive deviation analysis, repurposing resources like MAESTRO into high-level reasoning tasks suited for LLM evaluation. Alongside, we propose MuseAgent, a multimodal retrieval-augmented framework that integrates an LLM with perceptual front-ends: (i) a measure-wise OMR module producing symbolic representations (e.g., ABC notation Yuan et al. (2024a)), (ii) an AMT-based performance analysis module aligning audio with MusicXML scores and expressive JSON features, and (iii) a retrieval module supporting explicit and implicit access to symbolic/audio libraries. These modules ground the LLM in structured multimodal data, while a memory bank enables long-context, multi-turn reasoning. Together, MuseBench and MuseAgent provide the first foundation for advancing fine-grained multimodal music understanding.

Our evaluations on MuseBench reveal that general-purpose MLLMs demonstrate limited capabilities in handling symbolic music tasks, particularly when high-resolution scores or expressive audio are involved. While the text modality reflects the native performance of different base LLMs—with GPT-4.1 achieving the highest accuracy of 86.7%—MuseAgent exhibits clear advantages in the more challenging image and audio modalities. By leveraging specialized perceptual modules, MuseAgent achieves 74.1% accuracy on sheet image understanding and 88.1% on audio interpretation, significantly outperforming existing systems and validating the effectiveness of modular, multimodal reasoning for music understanding.

## 2 RELATED WORK

**Music Understanding Benchmarks.** Existing datasets Christodoulou et al. (2024); Hawthorne et al. (2019b); Li et al. (2018) for music research, such as MAESTRO Hawthorne et al. (2019b) and URMP Li et al. (2018), primarily focus on aligned score–audio pairs for automatic transcription or generation tasks. While these resources provide valuable training material, they lack task-oriented evaluation protocols and multimodal question-answering frameworks. MUSIC-AVQA Li et al. (2022)

and MuChoMusic Weck et al. (2024) introduce multimodal music Q&A datasets but do not focus on detailed symbolic score reading or nuanced performance interpretation. Our proposed *MuseBench* fills this gap by introducing a unified benchmark designed to assess Multimodal Large Language Models (MLLMs) across three interconnected abilities: text-based reasoning, score image interpretation, and audio-based performance analysis.

**Music Agents.**    Recent developments in general-purpose agents such as ReAct, Auto-GPT, and Gorilla have demonstrated strong capabilities in tool usage and reasoning. In the music domain, MuseNet Payne (2019) and MusicLM Agostinelli et al. (2023) focus mainly on symbolic or audio music generation. More recent efforts such as MusicAgent Yu et al. (2023) and AudioGPT Huang et al. (2024) incorporate external tools to process music inputs, showing early promise for intelligent music reasoning. However, none of these are designed specifically for understanding piano scores in conjunction with expressive performance audio. Our *MusicAgent* addresses this gap as the first agent tailored for piano music interpretation.

**Domain-Specific Multimodal Language Models.**    Inspired by successful domain-specific LLMs Luo et al. (2022); Huang et al. (2025); Manvi et al. (2024), several music-oriented models have recently emerged. MusiLingo Deng et al. (2024) targets music captioning and Q&A using instruction-tuned audio–language modeling. MuMu-LLaMA Liu et al. (2024) fuses music audio, images, and language using a unified LLM framework. NotaGPT Tang et al. (2025) was proposed as a large-scale visual language model specifically designed for music notation understanding. SymphonyNet Liu et al. (2022) demonstrates symbolic generation for orchestral music. These works affirm the value of domain-specific training and multimodal alignment, which we adopt in our MusicAgent to achieve more comprehensive understanding of piano music.

**General Multimodal Language Models.**    State-of-the-art general-purpose LLMs, including GPT-4o OpenAI (2023), Gemini Anil et al. (2023), Qwen Team (2024), and LLaVA Liu et al. (2023a), exhibit impressive performance on text–vision tasks. However, studies such as MuChoMusic Weck et al. (2024) reveal their limitations in music understanding, especially when tasks require interpreting structured music notation or expressive audio performance. These models often default to linguistic priors or hallucinate content in the absence of symbolic grounding, motivating the need for dedicated music-aware multimodal agents like ours.

**Distinctiveness of Our Work.**    In summary, existing benchmarks and agents for music understanding typically address only isolated modalities—such as symbolic scores or audio—or lack task-oriented evaluation frameworks. Most domain-specific or general multimodal models emphasize generation or vision–language alignment, but overlook the interplay between score reading and expressive performance. In contrast, our proposed MuseBench and MuseAgent jointly evaluate and interpret both symbolic scores and performance audio within a unified framework. This holistic design enables more nuanced assessment of multimodal reasoning in music and sets our work apart from prior research.

## 3 MUSEBENCH

To comprehensively evaluate the capabilities of multimodal large language models (MLLMs) in music understanding and analysis, we present **MuseBench**, a benchmark dataset that integrates **text**, **image**, and **audio** modalities. MuseBench combines diverse music data across these three modalities and assesses model performance through a rich set of multimodal tasks spanning multiple dimensions of music comprehension.

### 3.1 DATA SOURCES

To evaluate model performance in multimodal music understanding, we construct a high-quality dataset comprising sheet music images, real performance audio, and textual data. The dataset covers a wide variety of styles, eras, and difficulty levels, including Baroque, Classical, Romantic, and contemporary music. The overview of the dataset is shown in Figure 1, and further source details are provided in subsection A.1.

## 3.2 DATASET CONSTRUCTION

### 3.2.1 PREPROCESSING

We initially collected ∼3,000 candidate scores from multiple open repositories (see Appendix A.1). Scores were filtered for completeness, readability, and resolution quality. After normalization (resolution adjustment, background noise removal, and staff-line correction), ∼600 sheet images were retained. For audio, we collected 513 high-quality performance recordings. Each audio file was standardized to a uniform sampling rate and post-processed (denoising, normalization) to ensure clarity. To avoid copyright infringement, only recordings distributed under public licenses or explicitly provided by musicians with written consent were included.

### 3.2.2 ANNOTATION

Each sheet music image was paired with an ABC-format symbolic file containing metadata such as title, composer, key, time signature, note durations, and rhythm. Expert musicians further annotated technical difficulty and performance-related elements. Each piece was aligned with professional piano audio recordings and converted into MusicXML with bar-level score–audio alignment, forming a standardized metadata pool for subsequent task construction. All annotations were performed by trained musicians with at least five years of formal music education. To ensure reliability, multiple annotators cross-validated the labels, achieving a Cohen's $\kappa$ of 0.87.

### 3.2.3 DEFINITION

To evaluate the capabilities of MLLMs in music understanding and reasoning, we construct **MuseBench**, a benchmark collaboratively designed with expert musicians. It is structured around three core dimensions: (1) **music theory understanding**, (2) **sheet music understanding**, and (3) **performance audio analysis**. Each dimension contains multiple sub-tasks spanning diverse cognitive and perceptual challenges. Task definitions and evaluation criteria were established under expert consensus to ensure relevance and rigor.

A detailed description of each sub-task and its design rationale is provided in Appendix B. In total, we define **28 specific tasks** across the three modalities and six sub-dimensions, ensuring a balanced distribution of question–answer pairs and task formats. Our design draws inspiration from evaluation frameworks such as MMMU Yue et al. (2024) and OmniBench Li et al. (2024), while explicitly accounting for task difficulty and modality diversity to enable robust benchmarking.

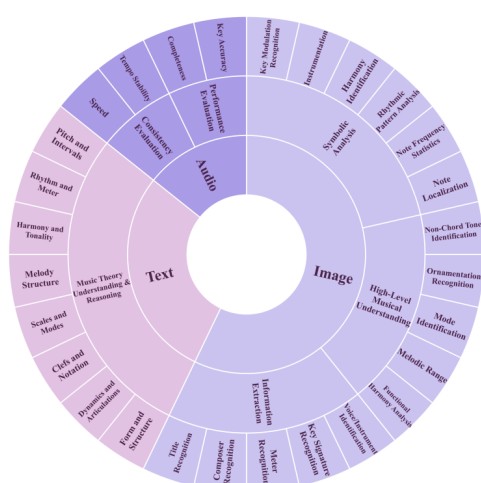

Figure 2: Distribution of questions in MuseBench. It consists of 28 task types across three modalities. Tasks are relatively evenly distributed to ensure balanced evaluation.

### 3.2.4 GENERATION

Based on the annotated metadata, we constructed multimodal question–answer pairs. Candidate questions were first generated using a combination of rule-based templates and GPT-4o OpenAI (2024) prompting to increase linguistic variety and naturalness. Ground-truth answers were derived deterministically from symbolic metadata through retrieval, calculation, and statistics, ensuring reproducibility. Expert musicians then reviewed and refined all question–answer pairs to guarantee correctness, clarity, and balanced difficulty.

The resulting dataset covers text, image, and audio modalities, offering comprehensive multimodal evaluation. Unlike prior datasets such as MusicTheoryBench Yuan et al. (2024b), which focus exclusively on text-based symbolic theory questions and include only a few hundred examples, MuseBench introduces multimodal QA tasks that demand joint reasoning over scores, audio, and

metadata. This process yields a large-scale benchmark comprising 591 sheet music images, 513 audio recordings, and 2,052 expert-verified question–answer pairs.

## 3.3 DATASET COMPLIANCE AND LICENSING

To ensure ethical and legal compliance, all components of MuseBench are sourced from either public-domain repositories (e.g., IMSLP, Mutopia, Project Gutenberg) or Creative Commons–licensed platforms (e.g., MuseScore), ensuring full legal compliance. Performance recordings are either public-domain or contributed with explicit consent under CC licenses. Further details on data sources, selection criteria, and license terms are provided in Appendix A.1.

Table 1: Defined tasks in MuseBench. The question format is randomly selected from a format pool for each task.The question types "MCQ," and "T/F" represent multiple-choice questions and judge true or false.

| Modality | Ability Dimension | Sub-task | Example Question | Type |
|---|---|---|---|---|
| Text | Music Theory Understanding & Reasoning | Pitch and Intervals | How many half steps are present in an augmented sixth interval? | MCQ |
| | | Rhythm and Meter | Which time signature represents a compound quadruple meter? | MCQ |
| | | Harmony and Tonality | Which pivot chord enables smooth modulation from C major to G major? | MCQ |
| | | Melody Structure | In a period structure, what describes the second phrase that resolves the first? | MCQ |
| | | Scales and Modes | The Dorian mode starting on D is derived from which major scale? | MCQ |
| | | Clefs and Notation | In alto clef, which pitch class is on the fourth space? | MCQ |
| | | Dynamics and Articulations | How should a musician perform staccato notes marked with a crescendo? | MCQ |
| | | Form and Structure | Which structural pattern best describes sonata-rondo form? | MCQ |
| Image | Information Extraction | Title Recognition | What is the title of the piece? | MCQ |
| | | Composer Recognition | Who is credited as the composer of the piece titled "Classical Rag"? | MCQ |
| | | Meter Recognition | What is the meter signature of the piece titled "The Waltz on my bum"? | MCQ |
| | | Key Signature Recognition | What is the key signature of the piece at the beginning of the score? | MCQ |
| | | Voice/Instrument Identification | Which voices are assigned to the bass clef? | MCQ |
| | Symbolic Analysis | Note Localization | In which measure does voice V:1 first play a chord containing the note E natural above middle C? | MCQ |
| | | Note Frequency Statistics | In voice V:3, which pitch class appears most frequently as a sounding note (excluding rests and grace notes) throughout the piece? | MCQ |
| | | Rhythmic Pattern Analysis | In the first four measures of the melody line (V:1), which rhythmic pattern is predominantly used for the repeated chord figures? | MCQ |
| | | Harmony Identification | In measure 18, which chord is formed by the combination of the soprano (V:1) and bass (V:3) notes? | MCQ |
| | | Instrumentation | Which staves in this score are written in the bass clef, indicating lower-pitched instrumentation? | MCQ |
| | High-Level Musical Understanding | Functional Harmony Analysis | What is the harmonic function of the raised A note $(^{a})$ $in the key of E-flat major as it appears in the melody line of V:1$? | MCQ |
| | | Mode Identification | Considering the key signature and the accidentals present throughout the melody in Voice 1, which mode is predominantly implied in this piece? | MCQ |
| | | Melodic Range | What is the melodic range (interval between the lowest and highest pitch) of the melody in voice V:1 throughout the piece? | MCQ |
| | | Ornamentation Recognition | In the melody line (V:1), which type of ornamentation is most consistently applied to the repeated patterns throughout the piece? | MCQ |
| | | Non-Chord Tone Identification | In measure 2 of Voice 1, which non-chord tone is most clearly functioning as a passing tone? | MCQ |
| | | Key Modulation Recognition | At which measure does the key signature effectively change from E-flat major to a key that includes an F-sharp accidental, indicating modulation? | MCQ |
| Audio | Performance Evaluation | Key Accuracy | Measure 11 confirms full key accuracy. | T/F |
| | | Completeness | Measure 8 has no missing notes. | T/F |
| | Consistency Evaluation | Tempo Stability | Measure 11's tempo matches reference stability. | T/F |
| | | Speed | Measure 8's speed is significantly slower than required. | T/F |

By integrating tasks across text, image, and audio modalities, the **MuseBench** dataset offers a comprehensive evaluation platform for multimodal large language models, spanning every facet of music understanding—from music theory comprehension to the assessment of real performance characteristics.

## 4 MUSEAGENT

We design MuseAgent as a multimodal retrieval-augmented large language model framework for music understanding and reasoning. Instead of treating perception and reasoning as a fixed pipeline, MuseAgent combines domain-specific perceptual modules with an LLM core in an agentic orchestration loop. Each perceptual module converts raw inputs from different modalities into structured symbolic representations—ABC notation from sheet images, JSON from performance audio, and MusicXML files from large-scale libraries. These structured representations form the basis of our retrieval-augmented generation (RAG) pipeline, enabling the LLM to ground its reasoning in domain-relevant symbolic and acoustic knowledge.

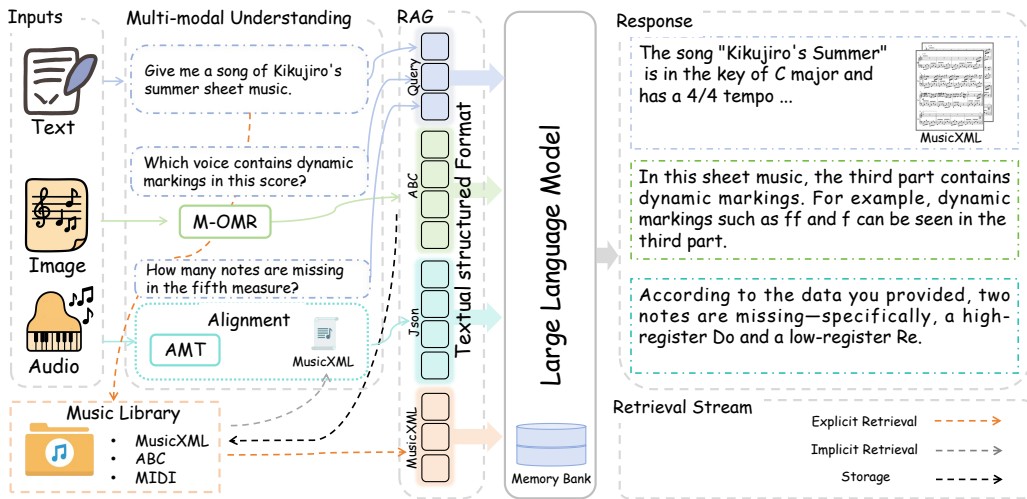

Figure 3: The MuseAgent framework integrates M-OMR, AMT, and music retrieval (explicit/implicit) into a unified large language model (LLM)-based system. Each perceptual module converts raw multimodal inputs into structured symbolic representations (e.g., ABC, MusicXML, JSON), which are incorporated into a retrieval-augmented generation (RAG) pipeline. The LLM acts as an agentic controller that dynamically orchestrates module usage depending on user intent, while a memory bank supports multi-turn dialogue and retrieval of prior outputs.

The framework design allows the LLM to dynamically orchestrate module usage depending on user intent. For instance, a query about harmony in a score triggers the measure-wise OMR module, while a question about tempo deviation invokes AMT-based alignment. Additionally, MuseAgent integrates a hybrid retrieval mechanism, supporting both explicit user queries (e.g., requesting a score) and implicit retrieval triggered internally by the LLM. A lightweight memory bank maintains intermediate outputs and conversation history, ensuring multi-turn, interactive dialogue across modalities.

## 4.1 MEASURE-WISE OPTICAL MUSIC RECOGNITION

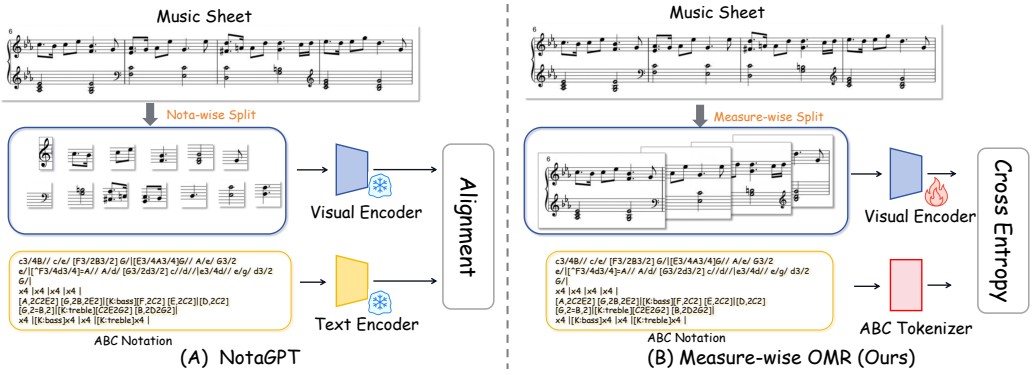

Figure 4: Comparison between (A) *NotaGPT*, which performs note-level segmentation with frozen visual and text encoders, and (B) our proposed *Measure-wise OMR* approach. The flame symbol denotes trainable modules, while the snowflake symbol indicates frozen components.

A key challenge for multimodal large language models (MLLMs) in music understanding is the modality gap between high-resolution score images and the symbolic reasoning required for musical analysis. Unlike natural images, music scores are densely structured and domain-specific, encoding hierarchical elements such as pitch, rhythm, and dynamics that general vision-language models struggle to interpret directly.

To address the challenge of structured music score recognition, we propose a Measure-wise Optical Music Recognition (M-OMR) module based on a "divide-and-combine" strategy. The input sheet

image is first divided into individual measures using visual layout cues such as staff lines and barlines. Each measure is treated as an independent visual unit, encoded into symbolic representation through localized recognition. These measure-level outputs are then combined to reconstruct the complete musical piece. The final result is expressed in ABC notation, a compact and structured symbolic format well-suited for downstream language model processing.

Unlike prior OMR approach, NotaGPT Tang et al. (2025), which split score images at the note level and rely on frozen vision and text encoders, we adopt a measure-wise segmentation strategy. The different between them as shown in Figure 4. By treating each measure as a semantic unit, our model preserves musical structure and reduces noise from overly granular splitting. We train a ResNet-based He et al. (2016) visual encoder, jointly trained with an LSTM Yu et al. (2019) over the measure sequence to capture intra-score dependencies. Furthermore, we introduce a custom-designed ABC tokenizer tailored for ABC-notation representation. This tokenizer captures over hundreds of ABC-notation variants of music-specific constructs (e.g., key, meter, chords), producing more compact and structurally meaningful token sequences compared to general-purpose text encoders.

This M-OMR module is integrated into the MuseAgent framework and significantly enhances its ability to interpret symbolic music. By leveraging the structured nature of scores and aligning recognized symbols with rhythmic metadata, the system achieves robust symbolic parsing suitable for downstream reasoning tasks. More details are shown in Appendix C.

## 4.2 AMT AND ALIGNMENT

To understand expressive performance audio, MuseAgent incorporates an Automatic Music Transcription (AMT) module and an audio-to-score alignment component. The AMT module transcribes raw audio into a symbolic representation (e.g., MusicXML) by extracting time-frequency features via the Constant-Q Transform Schörkhuber & Klapuri (2010), and applying neural transcription models.

The resulting symbolic sequence is temporally aligned to a reference score using a hierarchical Hidden Markov Model (H-HMM) Nakamura et al. (2015), which is robust to expressive timing variations, ornaments, and structural deviations such as repeats or skips. The alignment process produces structured outputs in JSON format, capturing onset timings, note correspondences, and expressive parameters.

These alignment outputs are then fused with user prompts and MusicXML files retrieved implicitly from the music library, forming the input to a retrieval-augmented generation (RAG) module. The RAG component composes these multimodal elements into an enriched prompt, enabling the language model to reason over both symbolic and auditory performance data. Implementation details, including model architecture and training configurations, are provided in Appendix D.

## 4.3 MUSIC RETRIEVAL MODULE

The MuseAgent supports both explicit and implicit retrieval from a large-scale symbolic music library in formats such as ABC, MusicXML, and MIDI.

For **explicit retrieval**, users can issue direct natural language queries (e.g., "Give me a song of *Kikujiro's Summer*") to fetch matching scores. For **implicit retrieval**, the system performs internal searches conditioned on audio, and sheet context, selecting relevant symbolic files (e.g., auio-paired MusicXML) to be integrated into the the RAG pipeline.

Unlike traditional information retrieval methods, retrieval here is embedded into the agent loop: the LLM may explicitly respond to user queries or implicitly call the retrieval API to ground its reasoning. This design realizes agentic RAG for multimodal music.

## 4.4 MUSIC THEORY UNDERSTANDING AND DIALOGUE CONTEXT

In addition to the aforementioned capabilities, MuseAgent harnesses the intrinsic musical knowledge embedded in large language models to answer music-theoretical questions (e.g., "What interval results from inverting a diminished fifth?" or "Which mode begins on E in the C major scale?"). The effectiveness of this ability may vary across used LLMs. For detailed evaluations of music theory understanding in different LLMs, please refer to Sec. 5.1.2.

The MuseAgent also supports memory capabilities for multi-turn conversations, for which we maintain a lightweight **memory bank** that stores intermediate module outputs, retrieved files, and previous model responses. The memory bank not only supports multi-turn reasoning, but also enables retrieval of prior structured outputs.

## 5 EXPERIMENT

### 5.1 COMPARISON ON MUSEBENCH

#### 5.1.1 BASELINE

We benchmark MuseAgent against 17 representative MLLMs spanning diverse categories to ensure a fair and comprehensive comparison: (i) **general-purpose LLMs** such as GPT-4.1, GLM-4, and Phi-4; (ii) **omni-modal models** including GPT-4o, Gemini 2.5-Pro, and Qwen2.5-Omni; (iii) **vision–language models** such as LLaVA, VisualGLM, and Qwen2.5-VL (7B/32B/72B); (iv) **audio-capable models** like Qwen2-audio and MuMu-LLaMA; and (v) **music-specialized models** including NotaGPT for notation understanding. This spectrum of baselines covers both generalist and domain-specific systems across text, image, and audio, allowing us to rigorously assess MuseAgent's advantages in music understanding.

#### 5.1.2 RESULTS ANALYSIS

Figure 5: MuseBench Performance Comparison. MuseAgent significantly improves image/audio tasks over omni-modal and domain-specific baselines.

| Modality | Model | Accuracy (%) |
|---|---|---|
| Text | GPT-4.1 OpenAI (2025) | **86.7** |
| | GPT-4o OpenAI (2023) | 85.5 |
| | Gemini2.5-Pro Comanici et al. (2025) | 83.9 |
| | GPT-4.1-mini OpenAI (2025) | 80.5 |
| | GPT-4.1-nano OpenAI (2025) | 73.3 |
| | GLM-4-PLUS GLM et al. (2024) | 78.2 |
| | GLM-4-FlashX GLM et al. (2024) | 60.3 |
| | Qwen2.5-72B Team (2024) | 81.4 |
| | Qwen2.5-32B Team (2024) | 79.8 |
| | Qwen2.5-7B Team (2024) | 71.3 |
| | Qwen2.5-Omni-7B Xu et al. (2025) | 63.3 |
| | Phi-4-14B Abdin et al. (2024) | 67.2 |
| | Random | 25.0 |
| Audio | MuseAgent (w/ GPT-4.1) | **88.1** |
| | MuseAgent (w/ GPT-4o-mini) | 78.9 |
| | MuseAgent (w/ GLM-4-FlashX) | 77.2 |
| | MuseAgent (w/ GPT-4.1-Nano) | 63.9 |
| | GPT-4o OpenAI (2024) | 55.9 |
| | Gemini2.5-Pro Comanici et al. (2025) | 53.1 |
| | MuMu-LLaMA Liu et al. (2024) | 51.7 |
| | Qwen2-audio Team (2024) | 51.4 |
| | Qwen2.5-Omni-7B Xu et al. (2025) | 50.6 |
| | Random | 50.0 |
| Image | MuseAgent (w/ GPT-4.1) | **74.1** |
| | MuseAgent (w/ GLM-4-FlashX) | 72.7 |
| | NotaGPT-7B Tang et al. (2025) | 68.1 |
| | GPT-4.1 OpenAI (2025) | 66.1 |
| | GPT-4o OpenAI (2024) | 64.2 |
| | GPT-4.1-mini OpenAI (2025) | 54.8 |
| | Gemini2.5-Pro Comanici et al. (2025) | 62.1 |
| | Qwen2.5-VL-72B Team (2024) | 58.9 |
| | Qwen2.5-VL-32B Team (2024) | 55.7 |
| | Qwen2.5-Omni-7B Xu et al. (2025) | 44.6 |
| | LLaVA-v1.5-13B Liu et al. (2023a) | 38.9 |
| | GLM-4V-9B GLM et al. (2024) | 37.1 |
| | Random | 25.0 |

**Text Modality:** GPT-4.1 achieves the highest textual accuracy (86.7%), slightly ahead of GPT-4o (85.5%) and Gemini 2.5-Pro (83.9). Larger models such as Qwen2.5-72B (81.4) and 32B (79.8) show improvements, yet still fall short.

**Audio Modality:** General-purpose omni models remain weak, with GPT-4o (55.9%) and Gemini 2.5-Pro (53.1%) close to random, and Qwen2.5-Omni (50.6%) performing similarly. Even Qwen2-audio, a specialized audio model, achieves only 51.4%. In contrast, MuseAgent with GPT-4.1 reaches 88.1%, highlighting the necessity of AMT and alignment modules for fine-grained performance analysis.

**Image Modality:** Vision–language models such as LLaVA (38.9%) and GLM-4V (37.1%) struggle with symbolic notation. Larger omni models like GPT-4o (64.2%) and Gemini 2.5-Pro (62.1%) perform better, and GPT-4.1 alone achieves 66.1%. However, integrating M-OMR pushes accuracy to 74.1%, surpassing both generalist models and music-specific baselines like NotaGPT (68.1).

**Key Insight:** Across modalities, neither model scaling (e.g., Qwen2.5-72B) nor omni-modal design (e.g., GPT-4o, Gemini 2.5-Pro) closes the gap in symbolic or performance-level reasoning. MuseAgent's modular design—combining M-OMR and AMT with LLM reasoning—consistently achieves state-of-the-art results, underscoring the need for domain-specific perceptual modules.

## 5.2 PERFORMANCE EVALUATION OF M-OMR AGAINST VLMs

We further benchmark our proposed M-OMR module against state-of-the-art Visual Language Models (VLMs) to evaluate its effectiveness in interpreting music scores. Following the evaluation protocol of NotaGPT Tang et al. (2025), we consider two tasks: (i) **closed-set** conversion of sheet music to ABC notation, evaluated by Levenshtein Distance, and (ii) **open-set** music analysis from visual scores, assessed by semantic metrics including LSA, ROUGE, and METEOR. This standardized setup ensures comparability with existing models.

Figure 6: Image-to-ABC Conversion Comparison. Results are evaluated on the standardized benchmark introduced in NotaGPT Tang et al. (2025).

| Model | Levenshtein Distance |
|---|---|
| VisualGLM-6B | 643.72 |
| DeepSeek-VL-7B-Chat | 308.27 |
| LLaVA-v1.5-13B | 147.47 |
| LLaVA-v1.6-Vicuna-13B | 918.94 |
| Qwen-VL | 439.82 |
| NotaGPT-7B | 59.47 |
| Gemini-pro-vision | 354.30 |
| GPT-4V | 655.45 |
| M-OMR (ours) | **18.39** |

Figure 7: Comparisons of open-source models and API-based models.

| Model | LSA | ROUGE-1 | ROUGE-L | METEOR | Avg |
|---|---|---|---|---|---|
| InternVL-Chat-v1.5 | 14.96 | 19.71 | 13.32 | 19.68 | 16.92 |
| VisualGLM-6B | 10.36 | 21.61 | 13.21 | 18.19 | 15.84 |
| DeepSeek-VL-7B-base | 9.92 | 16.43 | 11.60 | 13.81 | 12.94 |
| InstructBLIP-Vicuna-7B | 8.28 | 22.23 | 14.93 | 16.74 | 15.55 |
| InstructBLIP-Vicuna-13B | 8.37 | 20.29 | 14.18 | 14.17 | 14.25 |
| Qwen-VL | 9.58 | 15.21 | 10.37 | 12.56 | 11.93 |
| Qwen-VL-Chat | 9.66 | 16.80 | 11.37 | 14.42 | 13.06 |
| NotaGPT-7B | 12.46 | 22.63 | 15.53 | 18.34 | 17.24 |
| Gemini-pro-vision | 15.88 | 22.21 | 15.09 | **20.31** | 18.37 |
| GPT-4V | 14.03 | 18.49 | 11.36 | 19.94 | 15.96 |
| GPT-4o | **15.92** | 18.27 | 11.35 | 20.26 | 16.45 |
| MuseAgent (w/ M-OMR) | 15.75 | **24.92** | **15.76** | 20.17 | **19.15** |

### 5.2.1 CLOSED-SET IMAGE-TO-ABC NOTATION

We evaluate eight representative MLLMs, including API-based models (e.g., GPT-4V, Gemini Pro) and open-source models (e.g., LLaVA, VisualGLM, Qwen-VL-32B, NotaGPT-7B). As shown in Table 6, M-OMR achieves the lowest Levenshtein Distance (18.39), far surpassing all baselines such as NotaGPT-7B (59.47) and LLaVA-13B (147.47). These results demonstrate M-OMR's superior structural accuracy in symbolic notation conversion, highlighting its robustness in closed-set tasks.

### 5.2.2 OPEN-SET SCORE UNDERSTANDING

For open-set tasks, we compare models on semantic similarity and content relevance (Table 7). MuseAgent with M-OMR achieves the best average score (19.15), outperforming both strong API baselines such as GPT-4o (16.45) and Gemini Pro (18.37), as well as open-source vision–language models. Gains are consistent across metrics: higher ROUGE-1 and METEOR reflect better content coverage and fluency, while improved LSA highlights M-OMR's ability to capture nuanced musical semantics. Together, these results establish M-OMR as a robust and reliable solution for score interpretation within MuseAgent.

## 6 CONCLUSION

We introduced **MuseBench**, a comprehensive benchmark for multimodal music understanding, and **MuseAgent**, a modular agent that integrates symbolic score parsing and performance audio transcription. MuseBench spans 28 tasks across theory, score, and performance dimensions, offering a rigorous testbed for evaluating the reasoning capabilities of MLLMs. Experiments show that while general-purpose LLMs perform strongly on text-based tasks, they struggle with fine-grained score and audio understanding. By incorporating modality-specific modules such as M-OMR and AMT, MuseAgent achieves substantial gains in both image and audio modalities, demonstrating the necessity of domain-aware perceptual front-ends. These findings highlight the limits of pure scaling in generalist models and confirm the effectiveness of modular integration for complex music reasoning. We hope this work establishes a foundation for future research in AI-assisted music analysis, composition, and education, and for extending multimodal benchmarks beyond text, vision, and speech into the rich domain of music.

## ETHICS STATEMENT

We follow the ICLR Code of Ethics and take responsibility for all aspects of this work. All components of **MuseBench** are sourced from either public-domain repositories (e.g., IMSLP, Mutopia, Project Gutenberg) or Creative Commons–licensed platforms (e.g., MuseScore), ensuring full legal compliance. Performance recordings are either public-domain or contributed by professional musicians with explicit written consent. No copyrighted material or personal data are included. Annotations were conducted by qualified musicians who provided informed consent, and inter-annotator agreement was measured to ensure fairness and quality. We acknowledge that music understanding technologies may potentially be misused, e.g., for plagiarism or unauthorized content reproduction, and encourage users to apply MuseBench responsibly for research and educational purposes only.

## REPRODUCIBILITY STATEMENT

We have made extensive efforts to ensure the reproducibility of our work. All task definitions, preprocessing steps, and evaluation metrics are described in detail in the main text and Appendix B–E. Ground-truth answers are deterministically derived from symbolic metadata to ensure verifiability. Hyperparameters, training settings, and implementation details are documented in the appendix. Upon acceptance, we will release the full dataset (with license metadata), annotations, and codebase under a CC BY-NC 4.0 license.

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

## A  DATASET DETAILS

### A.1  DATA SOURCES AND SELECTION CRITERIA

To ensure both diversity and legal compliance, we selected music scores that are either (i) public domain works (composers deceased before 1954), or (ii) explicitly released under open licenses such as Creative Commons. No copyrighted material from the last 75 years was included. All recordings

either originated from public-domain datasets or were contributed by professional musicians with signed consent agreements.

The dataset includes:

- **Sheet Music Images:**
    - **MuseScore** MuseScore Team: ∼300 community-created scores (released under CC licenses), including modern and popular music.
    - **IMSLP** IMSLP / Petrucci Music Library: ∼200 high-resolution classical scores spanning 1600–1920, guaranteed public domain.
    - **Mutopia Project & Project Gutenberg** The Mutopia Project: ∼100 scores with public licenses, covering canonical works by Bach, Mozart, Beethoven, Chopin.
- **Textual Descriptions:** Metadata for each score includes title, composer, key, meter, and rhythmic structure, automatically extracted from symbolic files and verified by expert annotators.
- **Audio Files:** 513 piano performance recordings across classical, modern, and popular genres. All recordings are either public-domain (older archive sources) or provided directly by performers under Creative Commons licenses.

## A.2 BENCHMARK LICENSE AND USAGE

All components of **MuseBench** are released under strict legal and ethical compliance:

- **Sheet music** is drawn exclusively from public-domain repositories (IMSLP, Mutopia, Project Gutenberg) or from MuseScore where contributors licensed works under Creative Commons. Only works by composers deceased before 1954 are included.
- **Audio recordings** originate from public-domain archives or were directly contributed by professional musicians under written consent and Creative Commons licenses. No copyrighted recordings from the last 75 years are included.
- **Annotations** (ABC, MusicXML, task prompts) were prepared by trained musicians. Annotators provided informed consent, and inter-annotator agreement reached $\kappa = 0.87$.

**License:** MuseBench is released for non-commercial research and educational purposes under the **CC BY-NC 4.0** license. Redistribution or reuse of individual scores or recordings must comply with the original source licenses. Upon acceptance, we will publicly release all data, annotations, and evaluation scripts.

## B   DETAILED TASK DEFINITIONS

**Music Theory Understanding.**   This dimension focuses on textual comprehension of symbolic and conceptual music knowledge. It includes two sub-tasks:

- **Music Theory Recognition:** Evaluates understanding of basic music theory concepts, including key signatures, time signatures, note durations, and rhythmic structures.
- **Music Theory Reasoning:** Involves inferential questions that require deeper reasoning over symbolic descriptions of music, such as determining harmonic progression or identifying musical forms.

**Sheet Music Understanding.**   This dimension assesses the model's ability to interpret notated music from sheet images, and includes:

- **Information Extraction:** Transcription of basic musical metadata such as clefs, key signatures, and tempo markings from visual inputs.
- **Symbolic Analysis:** Understanding note symbols, their spatial and rhythmic relationships, and staff-based structural elements.
- **High-Level Interpretation:** Analyzing expressive or stylistic cues, such as articulation, phrasing, and functional roles in the musical context.

**Performance Audio Analysis.** This dimension assesses the model's ability to analyze expressive and structural characteristics in real performance recordings. It includes:

- **Performance Evaluation:** Judging the accuracy and completeness of a musical performance, including rhythmic precision, dynamic variation, and articulation clarity.
- **Consistency Evaluation:** Analyzing temporal stability, pitch consistency, and smoothness in expressive transitions across the performance.

## C    IMPLEMENTATION DETAILS OF THE M-OMR

The M-OMR module bases on a "divide-and-combine" strategy that serves as a visual encoder specialized for music score.

**Divide.** The input score image is initially segmented into individual measures through a combination of staff line detection and barline localization. Each segmented measure is then treated as an independent visual unit for localized recognition. Specifically, a YOLOv8-based detector Varghese & Sambath (2024) is employed to identify and localize each measure,

**Process.** Each measured image is encoded into a high-dimensional embedding using a ResNet-50 backbone He et al. (2016), capturing fine-grained visual features of musical symbols, including clefs, staves, barlines, key signatures, and time signatures. These embeddings are then sequentially decoded into ABC-format symbolic sequences using an LSTM-based decoder Yu et al. (2019) trained for note-level transcription.

**Combine.** The measure-level symbolic sequences are aggregated to reconstruct the full musical piece. During this step, time signatures extracted during pre-processing are aligned with each measure to ensure consistent rhythmic context. The final output is a well-formed ABC representation that preserves both temporal structure and notational correctness.

Recent studies have demonstrated the effectiveness of YOLO-based models in structured document analysis tasks Zhao et al. (2024).

Then, each segmented measure image is passed through a ResNet-50 He et al. (2016) encoder to obtain a latent visual embedding $\mathbf{x}_t$. The decoder is implemented as a unidirectional LSTM, which autoregressively generates the corresponding ABC sequence token-by-token.

$$\mathbf{h}_t = \text{LSTM}(\mathbf{h}_{t-1}, \mathbf{x}_t; \theta), \tag{1}$$

After decoding all measures, the symbolic output is reconstructed via:

$$\text{ABC}_{\text{full}} = \text{ConcatMeasures} \left\{ (\text{ABC}_i, \text{TimeSig}_i) \right\}_{i=1}^n, \tag{2}$$

where $\text{TimeSig}_i$ is the pre-detected time signature of measure $i$, and $n$ is the total number of measures.

**Datasets.** To construct a robust optical music recognition (OMR) module, we curated a large-scale dataset derived from the MuseScore platform, comprising over 80,000 music scores. Each score was first converted from MusicXML format to ABC notation Walshaw (2021), and subsequently rendered into SVG images. To further increase data diversity and model robustness, we performed structured data augmentation by randomly shuffling and replacing ABC bars, resulting in a synthetic corpus of 2.3 million ABC samples. Following image generation, we employed YOLO-based Varghese & Sambath (2024) segmentation to automatically detect and extract individual bars from the SVGs, ultimately yielding over **10 million** of image-bar pairs.

**Training Configurations.** The training was conducted over 100 epochs using a batch size of 12 and a learining rate of 1e-4. Our model achieved near accuracy (approximately 98%) on our held-out validation set, demonstrating both the scale and effectiveness of our training pipeline.

The visualization samples of ABC notation can be found in Figure 8. From the figure, we observe that MuseAgent, equipped with the M-OMR module, is able to accurately transcribe the entire sheet music into ABC notation. In contrast, other large language models struggle to extract complete and precise ABC representations, often missing structural or symbolic details.

## D IMPLEMENTATION DETAILS OF AMT AND AUDIO-TO-SCORE ALIGNMENT

This appendix provides the detailed implementation of the Automatic Music Transcription (AMT) and Audio-to-Score Alignment modules used in MuseAgent. While we adopt methods inspired by prior work Hawthorne et al. (2017); Nakamura et al. (2015), we report all relevant architecture and configuration details to facilitate reproducibility and downstream integration.

### D.1 AUTOMATIC MUSIC TRANSCRIPTION (AMT)

**Input Representation.**   We use the constant-Q transform (CQT) to extract time-frequency features from raw audio. A CNN–BiLSTM Siami-Namini et al. (2019) architecture predicts both onset and frame-level note activations, following the structure of Onsets-and-Frames Hawthorne et al. (2017).

Formally, given input features $\mathbf{X}_t$, the onset probability is predicted as:

$$\mathbf{O}_t = \sigma(\text{BiLSTM}(\text{CNN}(\mathbf{X}_t))), \tag{3}$$

and the framewise activation is computed as:

$$\mathbf{F}_t = \sigma(\text{BiLSTM}([\text{CNN}(\mathbf{X}_t), \mathbf{O}_t])). \tag{4}$$

The parameters are listed in Table 2.

Table 2: CQT Configuration for AMT

| Parameter | Value |
| --- | --- |
| Sample Rate | 16 kHz |
| Hop Length | 512 samples |
| Frequency Bins | 88 (covering A0–C8) |
| Bins per Octave | 12 |
| Window Function | Hann |
| Normalization | Log-magnitude |

**Network Architecture.**   The AMT model processes the CQT input through:

- **CNN Frontend:** 3 convolutional layers (kernel size: $3 \times 3$, stride: 1, padding: 1), each followed by ReLU and batch normalization.
- **BiLSTM Layer:** One bidirectional LSTM with 128 hidden units per direction.
- **Onset Head:** Fully connected layer with sigmoid activation to predict per-frame note onsets.
- **Frame Head:** Similar layer conditioned on onset features, used to predict framewise note activations.

**Training Details.**

- **Loss:** Binary cross-entropy loss applied independently to onset and frame predictions.
- **Optimizer:** Adam with learning rate $1 \times 10^{-4}$.
- **Training Epochs:** 50 on MAESTRO-V3 Hawthorne et al. (2019a).
- **Batch Size:** 8.

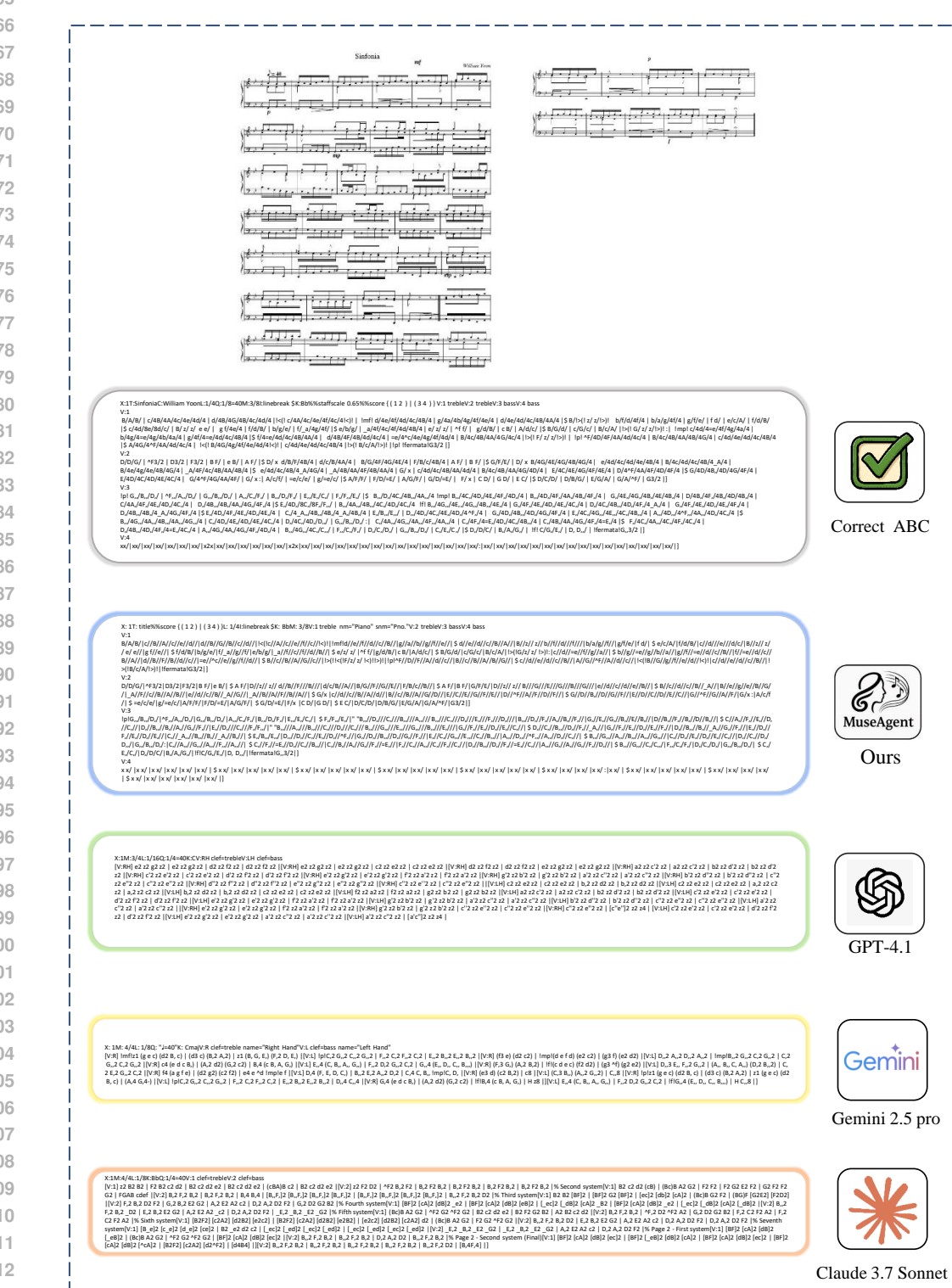

Figure 8: Performance of different LLMs on converting sheet music images to ABC notation

**Post-Processing.** Binary predictions are thresholded at 0.5. Onset and frame activations are merged into MIDI note events using the following heuristic: - A note onset is declared if the onset activation exceeds the threshold. - Note duration is extended over consecutive frames with active predictions.

The final output is exported in either MIDI or MusicXML format. We further convert to ABC notation when needed for symbolic alignment.

## D.2 AUDIO-TO-SCORE ALIGNMENT

**Model Overview.** We adopt the dual-layer HMM approach from Nakamura et al. (2015), which allows robust alignment between symbolic scores and AMT-derived audio events. The alignment process maximizes the posterior probability of the score position $p_t$ given observed acoustic features $x_t$:

$$p(p_t \mid x_t) = \frac{p(x_t \mid p_t) \cdot p(p_t)}{p(x_t)}. \tag{5}$$

**Structure.**

- **Top-layer HMM:** Models transitions between score positions (e.g., measures or note groups).
- **Bottom-layer HMM:** Captures fine-grained temporal dynamics within a note (onset, sustain, silence).

**Observation Model.** The likelihood $p(x_t \mid p_t)$ of observing acoustic feature $x_t$ given score position $p_t$ is modeled by a Gaussian Mixture Model (GMM):

- Number of components: 8
- Covariance: Diagonal
- Input: PCA-reduced CQT (dimension = 30)
- Training: Expectation-Maximization on aligned score–audio pairs

**Transition Model.** We define a transition matrix $A$ that supports:

- **Self-loop:** Sustains the current note position.
- **Forward transition:** Normal sequential progression.
- **Backward jump:** Repeat sections or corrections.
- **Forward skip:** Skipping sections.

These transitions are encoded as probabilities:

$$A_{ij} = p(p_t = j \mid p_{t-1} = i), \quad \text{with non-zero mass for } |i - j| > 1.$$

**Inference.** We use Viterbi decoding to compute the most probable alignment path:

$$p_{1:T}^* = \arg\max_{p_{1:T}} \prod_{t=1}^{T} p(x_t \mid p_t) \cdot A_{p_{t-1},p_t}$$

This algorithm effectively handles expressive timing, omission, repetition, and incorrect notes, making it robust for alignment in real-world performance scenarios.

## D.3 EVALUATION RESULTS OF AMT AND ALIGNMENT ALGORITHMS

This section presents the evaluation results of the Automatic Music Transcription (AMT) and alignment algorithms, which are recorded in a JSON format. These results provide a detailed assessment of various transcription metrics, such as overall accuracy, note matching, speed, stability, and tempo synchronization.

| Metric | Value |
|---|---|
| Overall Evaluation (eva_all) | 0.9252619743347168 |
| Note Evaluation (eva_note) | 1.0 |
| Speed Evaluation (eva_speed) | 1.0 |
| Stability Evaluation (eva_stability) | 0.7282252907752991 |
| Tempo Synchronization (eva_tempo_sync) | 1.0 |
| Extra Notes Count | 0 |
| Matched Notes Count | 2 |
| Missing Notes Count | 0 |

Table 3: Evaluation Results for Measure 37 of a Random Sample

The evaluation results for measure 37 of a random sample are summarized in the table below:

Additionally, the following figure 9 provides a visual representation of the performance comparison across the various transcription metrics.

## E EVALUATION DETAILS

### E.1 EVALUATION DETAILS IN OUR BENCHMARK

We present the different prompts used for three modalities: text, image, and audio. The following table summarizes the specific prompts for each modality.

Table 4: Prompts used for evaluation in our benchmark. The `<measure_id>` represents the unique identifier for each musical measure or section in question.

| Modality | System Prompt |
|---|---|
| Text | You are a music expert. Please read the following question carefully and provide the correct answer based on your knowledge of music theory and practice. |
| Image | You are a music expert. Please analyze the given sheet music image and select the correct answer to the question based on its notated content. |
| Audio | You are a music expert. Please carefully listen to the `<measure_id>` section of the provided audio excerpt and answer the question based on your auditory analysis. |

### E.2 EVALUATION METRICS USED IN CONTRAST EXPERIMENT

In this appendix, we present the evaluation metrics used in our M-OMR, comparing it with different models for converting images to ABC notation text, utilizing levenshtein distance. Additionally, we analyze music content using semantic similarity and word Matching metrics.

**Levenshtein Distance.** The Levenshtein Distance Yujian & Bo (2007) is used as the evaluation metric for converting images to ABC notation text. It refers to the minimum number of single-character operations required to transform model responses into the correct answer sequence.

Let $D$ be a matrix of size $(|R| + 1) \times (|A| + 1)$, where $|R|$ and $|A|$ represent the lengths of the response and answer sequences, respectively. $D[i][j]$ denotes the minimum edit distance between the first $i$ characters of $R$ and the first $j$ characters of $A$.

The subsequent values of $D$ are computed using the following recurrence relation:

$$D[i][j] = \min \begin{cases} D[i-1][j] + 1 & \text{(delete)} \\ D[i][j-1] + 1 & \text{(insert)} \\ D[i-1][j-1] + \text{cost} & \text{(substitute)} \end{cases}$$

where the cost is 0 if $R[i-1] = A[j-1]$, otherwise it is 1.

**Semantic Similarity and Word Matching Metrics.** Our Experiment also uses two categories of metrics: semantic similarity and word matching, for analyzing music content.

For semantic similarity, we use Latent Semantic Analysis (LSA), which measures the semantic similarity of text by computing the cosine similarity between vectors. The cosine similarity is given by:

## Measure 4

```
[
  {
    "avg_amplitude": 30.134309768676758,
    "avg_speed_ratio": 1.1799739599227905,
    "eva_all": 0.5308435559272766,
    "eva_all_ref": 0.6436206698417664,
    "eva_note": 0.2978394627571106,
    "eva_speed": 1.0,
    "eva_speed_ref": 0.9900000095367432,
    "eva_stability": 0.5706320405006409,
    "eva_stability_ref": 0.9900000095367432,
    "eva_tempo_sync": 1.0,
    "eva_tempo_sync_ref": 0.9900000095367432,
    "extra_note_count": 0,
    "id": 4,
    "matched_note_count": 1,
    "missing_note_count": 1,
    "neigh_note_count": 0,
    "note_pairs": [
      {
        "note_score": {
          "correct_pressed": false,
          "note_id": 18,
          "offtime": 3.875,
          "offtime_perf": 6.666150687110059,
          "ontime": 3.751041748046875,
          "ontime_perf": 6.56,
          "pitch": 71,
          "state": "missing"
        },
        "type": "normal"
      },
      {
        "note_perf": {
          "ID": "40",
          "confidence": 0.8005128502845764,
          "correct_pressed": true,
          "eva_tempo": 1.0,
          "hand": 2,
          "offtime": 7.452,
          "offtime_score": 4.62758622502449,
          "ontime": 6.56,
          "ontime_score": 3.751041748046875,
          "pitch": 74,
          "state": "matched",
          "vel": 90
        },
        "note_score": {
          "correct_pressed": true,
          "note_id": 19,
          "offtime": 3.875,
          "offtime_perf": 6.666150687110059,
          "ontime": 3.751041748046875,
          "ontime_perf": 6.56,
          "pitch": 74,
          "state": "matched"
        },
        "type": "normal"
      }
    ],
    "offtime_perf": 6.773999920114875,
    "offtime_perf_complete": 6.782999999952502,
    "offtime_score": 4.000941748049401,
    "ontime_perf": 6.56,
    "ontime_perf_complete": 6.56,
    "ontime_score": 3.751041748046875,
    "order": 4,
    "order_in_score": 4,
    "speed_ratio": 0.8563422560691833
  }
]
```

## Measure 37

```
[
  {
    "avg_amplitude": 30.134309768676758,
    "avg_speed_ratio": 1.1799739599227905,
    "eva_all": 0.9252619743347168,
    "eva_all_ref": 0.9939959645271301,
    "eva_note": 1.0,
    "eva_speed": 1.0,
    "eva_speed_ref": 0.9900000095367432,
    "eva_stability": 0.7282252907752991,
    "eva_stability_ref": 0.9900000095367432,
    "eva_tempo_sync": 1.0,
    "eva_tempo_sync_ref": 0.9900000095367432,
    "extra_note_count": 0,
    "id": 37,
    "matched_note_count": 2,
    "missing_note_count": 0,
    "neigh_note_count": 0,
    "note_pairs": [
      {
        "note_perf": {
          "ID": "503",
          "confidence": 0.8430560827255249,
          "correct_pressed": true,
          "eva_tempo": 1.0,
          "hand": 2,
          "offtime": 48.123999999999995,
          "offtime_score": 35.90951676290195,
          "ontime": 47.968,
          "ontime_score": 35.75104296875,
          "pitch": 71,
          "state": "matched",
          "vel": 90
        },
        "note_score": {
          "correct_pressed": true,
          "note_id": 197,
          "offtime": 35.875,
          "offtime_perf": 48.09002204596765,
          "ontime": 35.75104296875,
          "ontime_perf": 47.968,
          "pitch": 71,
          "state": "matched"
        },
        "type": "normal"
      },
      {
        "note_perf": {
          "ID": "504",
          "confidence": 0.8828990459442139,
          "correct_pressed": true,
          "eva_tempo": 1.0,
          "hand": 2,
          "offtime": 48.86,
          "offtime_score": 36.63806684602863,
          "ontime": 47.968,
          "ontime_score": 35.75104296875,
          "pitch": 74,
          "state": "matched",
          "vel": 90
        },
        "note_score": {
          "correct_pressed": true,
          "note_id": 198,
          "offtime": 35.875,
          "offtime_perf": 48.09002204596765,
          "ontime": 35.75104296875,
          "ontime_perf": 47.968,
          "pitch": 74,
          "state": "matched"
        },
        "type": "normal"
      }
    ],
    "offtime_perf": 48.21399902366102,
    "offtime_perf_complete": 48.2229999999525,
    "offtime_score": 36.000942968752526,
    "ontime_perf": 47.968,
    "ontime_perf_complete": 47.968,
    "ontime_score": 35.75104296875,
    "order": 37,
    "order_in_score": 37,
    "speed_ratio": 0.9843898415565491
  }
]
```

Figure 9: Results of Performance Comparison Across Transcription Metrics

For word matching, we use the following metrics:

- **ROUGE-1**: Calculates the number of unigram matches between the generated and reference text.
- **ROUGE-L**: Measures the longest common subsequence (LCS) match between the generated and reference text.
- **METEOR**: Calculates synonym matches and uses a combination of unigram matches, longest common subsequences, and synonym matches.

## F  LIMITATIONS AND FUTUREWORK

Currently, our benchmark and model primarily focus on tasks related to the piano, this specialization reflects a deliberate design decision—motivated by the availability of abundant piano data and the standardized nature of piano notation. Nevertheless, both MuseBench and MuseAgent are readily extensible to non-piano domains, such as guitar tablature, orchestral full scores, or improvisational jazz audio, given sufficient task-specific data. The modularity of our approach ensures that most components can generalize across musical styles with minimal adaptation. Expanding coverage to more diverse instruments and notational formats is a key direction for future work.

In addition, processing highly complex sheet images or long-form performance audio (e.g., tens of minutes) remains computationally demanding and may require further optimization. Finally, as with most LLM-based systems, the overall performance of MuseAgent is inherently constrained by the capabilities of the underlying language model and the quality of the input data.

## G  USE OF LLMs

We acknowledge the use of Large Language Models (LLMs) in the development of this work. Specifically:

- **Data Construction:** During benchmark generation, GPT-4o was employed to assist in drafting candidate natural-language questions based on annotated metadata. All corresponding ground-truth answers were derived deterministically from symbolic metadata (e.g., ABC, MusicXML) and subsequently verified and refined by expert musicians. LLM outputs were never used directly as final answers.
- **Writing Assistance:** LLMs (e.g., GPT-4o) were used for language polishing and paraphrasing of non-technical sections (e.g., abstract and introduction). All technical content, experiments, and results were designed and validated by the authors.

All uses of LLMs were supervised by the authors, and domain experts reviewed the outputs to ensure correctness, originality, and compliance with ethical guidelines. No parts of this paper rely solely on LLM output without human validation.

