# OpenReview forum: "Towards Multimodal Understanding of Music Scores and Performance Audio"
_ICLR.cc/2026/Conference — ICLR 2026 Conference Withdrawn Submission_

### Official Review · Reviewer_BnTL · 2025-10-22

**Soundness:** 2
**Presentation:** 2
**Contribution:** 2
**Rating:** 2
**Confidence:** 4

**Summary:**

The paper introduces MuseBench, a new benchmark for evaluating multimodal music understanding across text (theory), image (scores), and audio (performance). To address this benchmark, the authors propose MuseAgent, a framework that avoids end-to-end multimodal processing. Instead, it uses specialized perceptual modules: a Measure-wise Optical Music Recognition (M-OMR) module to convert score images into ABC notation (text) and an Automatic Music Transcription (AMT) module to convert performance audio into structured JSON (text). A large language model (LLM) then acts as an agent, using these structured text inputs and a RAG system to perform reasoning and answer questions. The central claim is that this modular, text-centric approach significantly outperforms current SOTA MLLMs, which fail at fine-grained score and audio understanding.

**Strengths:**

- The primary strength of this paper is the introduction of MuseBench. A new, three-modality benchmark that specifically tests fine-grained reasoning on scores and performance audio is a valuable contribution to the community. It provides a concrete testbed for evaluating MLLMs on a challenging, structured domain where they currently fall short.
- The paper is clearly written and identifies a real, well-defined problem: modern MLLMs (like GPT-4o and Gemini) are impressive generalists but fail at domain-specific, high-resolution perceptual tasks. The experiments clearly demonstrate this failure case (e.g., near-random performance on audio tasks).

**Weaknesses:**

- The paper's core finding that an LLM fed with specialized, structured text outperforms an MLLM fed with raw perceptual data is not a particularly surprising or novel conclusion. It's a known quantity that LLMs excel at symbolic reasoning when given clean, symbolic input. This "multimodal" solution feels more like a workaround that bypasses the core challenge of multimodal representation learning rather than solving it. It essentially converts a hard multimodal problem into an easier single-modality (NLP) problem.
- The innovation lies in the agentic pipeline and tool use (M-OMR, AMT) to feed a text-based LLM. This makes it a far better fit for an NLP conference (like ACL or EMNLP) that focuses on RAG, tool-use, and agentic reasoning.
- This is a major concern. The authors compare their M-OMR module's performance (Tables 6 & 7) against general-purpose Vision-Language Models (VLMs) like GPT-4V, LLaVA, and Gemini. This comparison is uninformative. To claim any novelty or state-of-the-art (SOTA) performance for M-OMR, it must be benchmarked against dedicated, SOTA OMR models from the Music Information Retrieval (MIR) community, which it fails to do.
- The OMR module directly targets ABC notation. This is a questionable design choice. ABC is a compact but structurally limited format. A more robust, standard, and likely higher-performing OMR pipeline would convert the score image to a richer symbolic representation like MusicXML or kern first, from which ABC notation can be losslessly derived. The current approach seems brittle and suboptimal.
- The paper's core finding confirms what is becoming increasingly apparent in MLLM evaluation: these models often default to text-based priors and symbolic reasoning, failing at true perceptual grounding. This has been shown in the audio domain (e.g., Zang et al., 2025 "Are you really listening? Boosting Perceptual Awareness in Music-QA Benchmarks") and in the vision domain (e.g., Chen et al., 2024, "Are We on the Right Way for Evaluating Large Vision-Language Models?"). This paper provides another example of this phenomenon but, by leaning into the text-only workaround, it doesn't solve the underlying MLLM problem.

**Questions:**

1. Why did you not compare your M-OMR module against any established SOTA OMR models? Can you provide benchmark results comparing M-OMR to models from the MIR literature on standard OMR datasets?
2. Could you justify the decision to pursue a direct image-to-ABC conversion? Have you benchmarked this approach against a more standard pipeline (e.g., Image $\rightarrow$ MusicXML $\rightarrow$ ABC), which would likely be more robust?
3. Prior work (e.g., Zang et al., 2024 in music; Chen et al., 2024 in vision) has already demonstrated that MLLM evaluations often test text-based priors rather than true multimodal perception. Your paper seems to confirm this by showing that a pure text-based pipeline (using your "translation" modules) is superior. How do you differentiate your contribution from simply providing an engineering workaround for this known limitation?

---

### Official Review · Reviewer_5d1N · 2025-10-26

**Soundness:** 2
**Presentation:** 3
**Contribution:** 3
**Rating:** 4
**Confidence:** 4

**Summary:**

This paper proposes a benchmark called MuseBench for multimodal understanding of music, covering image scores, performance audio, and text. In addition, a framework called MuseAgent is proposed which includes OMR and AMT components, which is able to outperform other general purpose MLLMs when evaluated against the proposed benchmark.

**Strengths:**

* Although there are several existing benchmarks for music understanding, they typically capture single or pairs of modalities - whereas MuseBench attempts to evaluate jointly both image scores and performance audio and as thus it stands apart from other efforts.
* The effort in collecting openly available data for images, scores, and music audio is commendable.

**Weaknesses:**

* Although it can be difficult to source audio and corresponding scores, one question might be whether having around 500 objects per modality is sufficient towards constructing a dataset and benchmark.
* There are several points across the manuscript where there is a lack of clarity. More details can be found in the Questions field below.
* It would appear that there are limited tasks covering the audio modality when compared with the text and image modalities. Audio tasks are only covered through T/F questions as well. This poses a question on whether this benchmark is capable of covering nuances of audio performance in the same level with capturing aspects of music composition and music theory.
* The paper's writing quality can be improved. A particular problem is that citations in the text are incorrectly formatted (authors should be careful when to use \cite versus \citep).

**Questions:**

* Throughout the manuscript, citations are incorrectly formatted - please check when to use \cite versus \citep in the paper.
* The "measure-wise OMR" contribution is unclear. Would that mean that the image score is segmented into measures manually or using metadata? Or is the agent able to automatically split the image score into measures which are then fed into the OMR component?
* Section 1 mentions "74.1 accuracy on sheet image understanding and 88.1% on audio interpretation". These measures are not clearly defined in the text, and there is a question on whether the MCQ and T/F questions are able to capture sheet image understanding and audio interpretation.
* From section 3.2, it is not clear whether the authors collected image scores only or collected underlying machine-readable scores (such as MusicXML). Were the 600 sheet images paired with the 513 audio performance recordings? From 3.2.2, it is unclear what the metadata information on "note durations" and "rhythm" includes (since a piece can easily have hundreds of notes). Was each piece manually aligned with professional piano audio recordings, or was the alignment automatic? (if yes to the latter, was the automatic alignment accurate?)
* From section 3.3, it appears that all tasks in MuseBench include MCQ and T/F questions (with the audio modality having significantly less numbers of tasks). One question is whether these tasks are truly able to assist with sheet image and audio understanding, versus more traditional tasks on OMR and AMT which produce detailed outputs?
* From section 5.2, it is unclear why the paper only includes results on OMR while other tasks are moved to the appendix.
* A.1: From MuseScore, did the authors download PDFs of image scores? Or machine-readable scores which were rendered into images?

---

### Official Review · Reviewer_X2wX · 2025-10-29

**Soundness:** 2
**Presentation:** 2
**Contribution:** 2
**Rating:** 4
**Confidence:** 3

**Summary:**

This paper introduces MuseBench, a benchmark for evaluating multimodal large language models (MLLMs) on symbolic–acoustic music understanding, and MuseAgent, an agent framework that integrates optical music recognition (M-OMR), automatic music transcription (AMT), and retrieval-augmented generation (RAG).
The goal is to enable cross-modal reasoning between sheet scores and performance audio, and to test how well LLMs can handle music theory, symbolic notation, and expressive performance.

The paper is technically comprehensive, with well-structured dataset construction and extensive baseline comparisons. However, it remains unclear what practical capabilities the system actually enables beyond benchmarking, and whether the proposed “agent” achieves genuine multimodal understanding rather than modular fusion of existing components.

**Strengths:**

Valuable benchmark contribution. MuseBench systematically organizes symbolic, audio, and textual reasoning tasks, which will likely be useful for future MLLM evaluation.

Well-engineered perceptual front-ends. The measure-wise OMR and AMT+alignment modules are carefully implemented, with clear quantitative improvements over existing systems.

Comprehensive comparison. The inclusion of a large spectrum of baselines helps position the results clearly in the multimodal LLM landscape.

**Weaknesses:**

**Unclear concrete capability**

The paper claims “multimodal understanding” but gives only static QA examples.
It is unclear what MuseAgent can actually do in practice:

    Can it compare a score and a recording, and highlight discrepancies (wrong notes, timing errors)?

    Can it detect improvisations or expressive deviations beyond symbolic correctness?

Without such demos, it is hard to judge whether this is a functional system or a benchmarking pipeline.
A short interactive or visual demo (e.g., highlighting mismatched notes in a score–audio pair) would greatly clarify real capabilities.

**Model independence and emergent behavior**

MuseAgent integrates LLM + OMR + AMT + HMM.. This raises the question: is the system itself capable of cross-modal reasoning, or is it a stitched combination of preexisting models? If the latter, it weakens the claim of an “agentic” -- for the current landscape of multimodal LLM research, people expect at least some “emergent” understanding. The authors should analyze what aspects require the LLM—e.g., can it reason about expressive intent, or does it merely aggregate module outputs? Otherwise, the contribution risks being a pipeline rather than a unified model.

**Ambiguous terminology and unclear task definition**

The terminology (e.g., “agent-based orchestration,” “understanding,” “RAG”) is broad and occasionally inconsistent with the presented experiments. Is “orchestration” only a metaphor for module calling? or can it orchestration of an input melody by RAG？ In general, clarifying this distinction between music reasoning, music retrieval, and music generation would make the scope much clearer.

**Questions:**

as indicated in the weakness

---

### Official Review · Reviewer_AwfY · 2025-11-01

**Soundness:** 4
**Presentation:** 3
**Contribution:** 4
**Rating:** 4
**Confidence:** 5

**Summary:**

This paper addresses a key limitation of current multimodal LLMs (MLLMs): their inability to perform joint, fine-grained reasoning over complex musical inputs—specifically sheet music (image) and performance audio (acoustic signal). The authors make two key contributions:
1. MuseBench: A novel tri-modal benchmark for rigorously evaluating MLLMs across fundamental music theory (text), score-based reasoning (image), and performance-level interpretation (audio).
2. MuseAgent: A modular RAG framework featuring two specialized perceptual modules—M-OMR (Measure-wise Optical Music Recognition) and AMT (Automatic Music Transcription) with H-HMM (Hierarchical Hidden Markov Model) alignment—to convert heterogeneous inputs into structured, LLM-interpretable text (e.g., ABC notation, JSON).
MuseBench reveals that general MLLMs (e.g., GPT-4o, Gemini 2.5-Pro) perform poorly on image/audio music tasks—near-random on audio—while MuseAgent’s modular, domain-specific design achieves SOTA results.

**Strengths:**

● High-Impact Problem and Novel Framework: The paper tackles a significant and timely challenge. The proposed solution—decoupling specialized perception (M-OMR, AMT) from general reasoning (LLM core)—is a practical and effective paradigm for integrating MLLMs into highly structured domains like music.
● Benchmark Construction (MuseBench): MuseBench is a major contribution in itself. It is the first benchmark to jointly evaluate score and performance audio understanding. The data construction is robust, involving expert verification (Cohen's $\kappa=0.87$) and deriving all ground-truth answers deterministically from symbolic metadata, ensuring the benchmark's quality and objectivity.
● Empirical Superiority of MuseAgent: MuseAgent’s specialized modules decisively outperform general MLLMs—74.1/88.1 % vs 68.1/55.9 % on image/audio tasks—thanks to M-OMR’s measure-wise segmentation (Levenshtein 18.39) and H-HMM robust audio-score alignment.

**Weaknesses:**

● Limited Ablation Study for M-OMR Components:  The specific contribution of the custom ABC tokenizer in the M-OMR module is not isolated from the contribution of the measure-wise segmentation in the ablation study.
● Generalization to Non-Piano Music: The authors acknowledge in the limitations section that the work primarily focuses on piano scores and state that the framework is "readily extensible" to other instruments. While plausible, this is a strong claim. The performance of the M-OMR's measure-wise segmentation in complex, multi-staff orchestral scores (which have different alignment challenges than piano systems) is questionable and warrants a more cautious or deeper discussion in the limitations section.
● Lack of Explicit Cross-Modal Interaction Tasks: The benchmark tasks are largely segregated (Image-only or Audio-only). MuseBench currently lacks tasks that mandatorily require the model to jointly process and compare both the raw score image and the raw performance audio for a complex answer.

**Questions:**

● Cross-Modal Task Design: Could the authors comment on the feasibility and design of a dedicated Score & Audio interaction task category for a future version of MuseBench, where neither modality alone is sufficient to answer the question?
● LLM Constraints: Given that the final performance is constrained by the underlying LLM, what specific types of reasoning errors did the LLM core make, even when provided with perfect structured input from M-OMR and AMT?
● Technical Details on H-HMM: For the AMT alignment module, could the authors briefly elaborate on the specific hidden states utilized in the Hierarchical HMM (H-HMM)? Specifically, how are performance errors (e.g., missed notes, spurious notes) modeled within the H-HMM's hierarchy?

---

### Note · Authors · 2026-01-06

I have read and agree with the venue's withdrawal policy on behalf of myself and my co-authors.